# The Endolysosomal System: The Acid Test for SARS-CoV-2

**DOI:** 10.3390/ijms23094576

**Published:** 2022-04-20

**Authors:** Daniella Cesar-Silva, Filipe S. Pereira-Dutra, Ana Lucia Moraes Giannini, Cecília Jacques G. de Almeida

**Affiliations:** 1Laboratory of Immunopharmacology, Oswaldo Cruz Institute, Fundação Oswaldo Cruz, Rio de Janeiro 21040-900, Brazil; daniellasilva@aluno.fiocruz.br (D.C.-S.); filipe.dutra@ioc.fiocruz.br (F.S.P.-D.); 2Laboratory of Functional Genomics and Signal Transduction, Universidade Federal do Rio de Janeiro, Rio de Janeiro 21941-901, Brazil; ana.giannini@ufrj.br

**Keywords:** lysosome, endosome, acidification, coronaviruses, endocytosis, autophagy, SARS-CoV-2, virus cycle, immune evasion, immunity

## Abstract

This review aims to describe and discuss the different functions of the endolysosomal system, from homeostasis to its vital role during viral infections. We will initially describe endolysosomal system’s main functions, presenting recent data on how its compartments are essential for host defense to explore later how SARS-CoV-2 (Severe Acute Respiratory Syndrome Coronavirus 2) and other coronaviruses subvert these organelles for their benefit. It is clear that to succeed, pathogens’ evolution favored the establishment of ways to avoid, escape, or manipulate lysosomal function. The unavoidable coexistence with such an unfriendly milieu imposed on viruses the establishment of a vast array of strategies to make the most out of the invaded cell’s machinery to produce new viruses and maneuvers to escape the host’s defense system.

## 1. The Endolysosomal System

Let us begin by briefly describing the organelles and processes this review will focus on.

### 1.1. Lysosomes

The lysosome is the cellular terminal station where the degradation of internalized material occurs. It performs a crucial role in keeping homeostasis, and defective function can lead to disease. These organelles store a vast repertoire of inactive acid hydrolases, activated upon fusion with the acidic late endosomes. This merge forms the endolysosomes, where degradation takes place [1]. It is important to emphasize that the lysosome term is commonly used to define the degradative compartment of the cell, but this definition is not accurate. The degradative compartments are the vesicles that originate from the fusion with lysosomes, such as endolysosomes, autolysosomes, and phagolysosomes (Figure 1). These coexist with terminal storage lysosomes that are not acidic and serve to keep acidic hydrolases that will be activated when needed upon fusion with cargo-containing acidic vesicles [1]. We will respect the nomenclature adopted in each cited article to construct this review. Still, the reader must be aware of the broad meaning the term lysosome may have in some articles, where it may be used as a synonym for endolysosomes. 

Lysosomes and the other vesicles of the endolysosomal system have their specificities, including cargo, resident proteins, membrane markers, morphology, redox environment, and pH. Vesicles resulting from the fusion of distinct compartments–such as amphisomes, autolysosomes, endolysosomes, phagolysosomes-exhibit hybrid characteristics from the parental compartments (Figure 1). Together, these features allow these organelles to accomplish their functions. The luminal acidic pH of degradative compartments can be as low as 4.5–5.0, and this is maintained by the V-type ATPase, and ion transporters and channels keep the ionic balance in the lumen. To shield the luminal lysosome membrane against the destructive potential of acid hydrolases, LAMPs (lysosome-associated membrane proteins) are highly glycosylated in their luminal N-portion [2]. Also, as mentioned before, these hydrolases are only fully activated when lysosomes and late endosomes fuse. After fusion, lysosomes are reformed by a maturation process that condensates its contents and recaptures endosomal-specific proteins [3].

### 1.2. Endocytosis and the Endolysosomal System

The endosomal system comprises vesicle compartments in the cell that regulate the traffic and fate of material from the cell itself or acquired after internalization. 

Endocytosis is the cellular process that allows the intake of fluid, solutes, macromolecules, and particles through membrane invagination. It may be constitutive or receptor-dependent, and in each case, the plasma membrane forms invaginations through distinct mechanisms, dependent on specific molecules and structures, such as clathrin, caveolae, Cdc42 (Cell Division Cycle 42), Rho/Rac, endophilin A2 toxins, and FEME. The variety of these mechanisms is not the scope of this review, and readers should refer to excellent reviews in the area, such as one of Sandvig and collaborators [4]. 

Following membrane invaginations, the newly formed vesicles fuse with early endosomes, where the mildly acidic pH (ranging from 5.9 to 6.8) provokes the dissociation of many ligands from their receptors. In endosomal compartments, ingested material to be destroyed (large particles such as viruses) and molecules to be recycled are sorted out. The early endosome is responsible for sorting and delivering distinct molecules to their destination compartments. This is achieved by the highly organized feature of these compartments where vacuolar sorting domains and tubular recycling domains are found. The first contains the material to be degraded and mature to late endosomes, while the later domains have a high surface membrane/lumen ratio to recycle molecules to other compartment membranes such as plasma membrane, TGN, and recycling endosomes, without losing nutrients. To achieve correct sorting, membrane subdomains of the endosome are distinctly enriched in Rab5, Rab4, Rab11, Arf1/COPI (ADP-Ribosylation Factor 1/Coat Protein I), retromer, and caveolae [5,6,7]. 

The Rab proteins-small GTPases (guanosine triphosphatases) of the Ras superfamily-are crucial for vesicle trafficking and fusion and participate in several steps of this process. For instance, Rab5, a component of early endosomes, regulates vesicle fusion and sorting of molecules and is lost in the maturing process into late endosomes, while Rab7 is acquired in this maturation. Rab7 has crucial roles in endolysosomes, regulating processes, such as lysosomal enzymes trafficking, fusion with lysosomes, vesicle acidification, and degradation [8]. In addition, Rab4 and Rab11 handle the recycling of membrane receptors to the plasma membrane via recycling endosomes [5,9,10].

Membrane fusion involves the coordinated interaction of Rab GTPases, SNAREs (soluble N-ethylmaleimide-sensitive factor attachment protein receptor), the ESCRT (endosomal sorting complexes required for transport) machinery, and tethering complexes-CORVET (class C core vacuole/endosome tethering) and HOPS (homotypic fusion and vacuole protein sorting). The SNARE family of proteins comprises 38 members, classified according to the presence of conserved glutamine (Q) or arginine (R) residues in R-SNARES or Q-SNARES (further distinguished in Qa-, Qb- and Qc-SNARE). The process of fusion depends on the interaction between SNARES of opposing membranes. SNARES assemble in a quadruple helix forming a zipper that pulls the membranes so close together that they fuse [11]. ESCRT is a multimolecular complex that drives membrane remodeling. It consists of four main complexes (ESCRT-0, -I, -II, and -III) and associated proteins. ESCRT-0 interacts with ubiquitylated proteins, recruiting ESCRT-I, -II -III in a step-wise manner. Tethering complexes CORVET and HOPS share a core comprising the Vps (Vacuolar Protein Sorting) proteins Vps11, Vps13, Vps18, and Vps33 associated with Vps3 and Vps8 in CORVET or Vps39 and Vps41 in HOPS. Whereas CORVET participates in fusing early endosomes to other vesicles, HOPS is a tethering complex involved in fusion between the lysosome and other vesicles [12,13,14].

In summary, as early endosomes mature to late endosomes, there is an exchange of GTPases, tethering complexes, and also SNAREs in a highly controlled mechanism that assures identity and proper functioning of each compartment. We will refer to specific members of these families of proteins along this review to describe events important for the coronaviruses cycle. 

Early endosomes also communicate with the Golgi, receiving macromolecules, such as acid hydrolases from the trans-Golgi network. These hydrolases may also be secreted by the cell through the secretory pathway and endocytosed back after they bind to mannose-6-phosphatase receptors (M6PR) present in the plasma membrane [15]. Thus, communication with other compartments is essential for the maturation of early endosomes into late endosomes, allowing the acquisition of distinct molecules and the delivery of others. Late endosomes pH ranges between 6.0–4.8, and the maturation of early into late endosomes is also characterized by the formation of inward budding of endosomal membrane creating intraluminal vesicles (ILVs) [6]. As the maturation process continues, ILVs accumulate, and in late endosomes, they are abundant and characterize multi-vesicular bodies (MVBs). MVBs can fuse to lysosomes, where ILVs are degraded, or fuse to the membrane liberating ILVs, constituting the exosomes [6]. 

MVBs are formed by two distinct mechanisms, one dependent and the other independent of ESCRT. The Vps4 provides the energy for dissociating the inward budding vesicles from the endosomal membrane. Fusion of MVBs with lysosomes heads the ubiquitin-tagged proteins to degradation [16,17]. An alternative mechanism relies on the activity of SMase (sphingomyelinases), enzymes exhibiting different optimal pHs (alkaline, neutral, or acid), that hydrolyses SM (sphingomyelin) of endosomal lipid rafts to ceramide. Plasma and endosomal membranes have an asymmetrical lipid composition, with SM enriched in the non-cytosolic leaflet. Whereas sphingomyelin has a cylinder topology and a high affinity for membrane cholesterol that contributes to the compact and rigid characteristic of lipid rafts, ceramide, with its conical topology, favors vesicle budding by influencing cholesterol efflux, membrane fluidity, and curving [18]. ESCRT-dependent and -independent pathways can be inhibited by different drugs, such as manumycin, an inhibitor of Ras farnesylation, and GW4869, an inhibitor of the nSMase (neutral SMase), respectively [17,19].

### 1.3. Lysosomes as Signaling Hubs: From Lysosomal Exocytosis to Autophagy Control

ILVS release from the cell during lysosomal exocytosis constitutes an essential way of cell communication [20]. Lysosome exocytosis is also involved in membrane repair, degradation, and remodeling of the extracellular matrix and release of non-degraded material. Thus, lysosomes not only act inside the cell but function beyond the limits of the plasma membrane. To accomplish all these tasks, lysosomes’ location needs to be highly dynamic, which is made possible by interaction with extensive microtubule network going either outwards using kinesin or inwards using dynein. 

The anterograde movement of lysosomes depends on the complex of proteins in the lysosomal membrane composed by BORC, the small GTPase Arl8, and the kinesin-binding protein PLEKHM2/SKIP (Pleckstrin homology domain-containing family M member 2/Salmonella-induced filaments A and kinesin-interacting protein) that connects lysosomes to kinesin [21,22,23,24]. Distinct GTPases are also implicated depending on the cell type, for example, Rab27a and Rab27b in HeLa cells, Rab11 in K562 cells, Rab35 in Oli-neu cells [17,25,26,27]. 

Lysosomes’ interaction with the actin cytoskeleton permits its appropriate positioning close to the membrane, which facilitates fusion between these two compartments. This event also requires mobilization of specific SNARES of both organelles, such as VAMP7 and SNAP23, which dock the lysosome to the internal face of the plasma membrane, respectively. Besides, fusion is also dependent on Ca^2+^ release from the lysosomal lumen through channels such as TMPRL1. The presence of Ca^2+^ triggers lysosomal synaptotagmin VII interaction with the Qb,c-SNARE SNAP23, and the Qa-SNARE Syntaxin 4, facilitating the formation of the SNARE complex with the R-SNARE VAMP7 [20,28].

The location of lysosomes is also associated with their function as a metabolic hub that integrates cell signals triggering or inhibiting anabolic/catabolic activity [29]. 

A key player in this setting is mTORC1 (mechanistic target of rapamycin complex 1)-a complex composed by mTOR (mechanistic target of rapamycin), and the adaptor proteins Raptor and MLST8 (Mammalian lethal with SEC13 protein 8/Target of rapamycin complex subunit LST8)-that in its active form localizes to the cytosolic membrane of late endosomes and lysosomes. Active mTORC1 senses hormonal cues and the metabolic status of the cell, promoting ribosomes biosynthesis, protein translation, lipogenesis while inhibiting autophagy. mTORC1 is activated by growth-factor-activated Rheb (Ras homolog enriched in brain) GTPase and by amino acids-activated Rag GTPases [30,31,32].

The protein synthesis promoting activity of Rheb is inhibited by the tuberous sclerosis complex (TSC). Growth factors liberate Rheb from TSC inhibition by activating PI3K and PKB/Akt1 and subsequent phosphorylation of TSC, which becomes sequestered in the cytoplasm, away from lysosomes. Amino acids activate Rag by stimulating its interaction with Raptor, bringing mTORC1 to late endosomes and lysosomal membranes, where mTORC1 interacts with Rheb. Interestingly, activation of mTORC1 by amino acids requires the activity of v-ATPase. The active mTORC1 phosphorylates substrates, such as S6 kinase (S6K1), eukaryotic translation initiation factor 4E-binding protein 1 (4E-BP1), Unc-51 like autophagy activating kinase (ULK1), and transcription factor EB (TFEB). Overall mTORC1 actions stimulate anabolic activity and suppress activators of autophagy. Lysosome positioning close to the plasma membrane can help enhance mTORC1 activation because activators such as Akt1 are close by [30,31,32].

During starvation, anabolic activities are inhibited, and autophagy of self-organelles and macromolecules is activated to provide nutrients for the cell. To this end, the biogenesis of lysosomes and autophagosomes is stimulated, as well as the fusion of these organelles. Autophagy begins with mobilization of Atg (autophagy-related gene) proteins to the phagophore assembly site, and nucleation of a cup-shaped membrane termed IM (isolation membrane). This structure forms the phagophore, which expands surrounding cargoes until it confines them in double-membrane vesicles, which characterize the autophagosome. The outer membrane of autophagosomes fuses with the membrane of lysosomes delivering its cargo for degradation in the newly formed autophagolysosome [33]. This process involves the coordinated recruitment of distinct complexes of proteins that we will describe briefly for understanding the mechanisms that coronaviruses manipulate for their own benefit, which will be depicted later in this review. 

In nutrient-rich conditions, mTORC1 interacts with the ULK1 (Unc-51 Like Autophagy Activating Kinase 1) complex, formed by ULK1 itself, Atg13, FIP200, and Atg101 (family kinase-interacting protein of 200 kD) and phosphorylates ULK1 at specific residues (serines 637 and 757 in mouse and serines 638 and 758 in human; the first is phosphorylated by AMPK and MTOR and the second solely by mTOR) [34,35,36,37], rendering ULK1 inactive. Starvation induces inactivation and moving of mTORC1 to the cytoplasm. The phosphorylation of ULK1 is then relieved by phosphatases, and ULK1 is autophosphorylated (at serine 1042 and threonines 180 and 1046) [38]. In this form, ULK1 becomes active, phosphorylates FIP200 (at serine 943, 986, and 1323) and Atg13 (at serines 203 and 318) [39,40], moves to the endoplasmic reticulum recruiting a second complex, termed the PI3KC3 (phosphatidylinositol 3-kinase catalytic subunit type 3) complex II. This complex consists of Vps34-a class III phosphatidylinositol 3-kinase, Beclin-1 (Bcl-2-interacting protein 1), Vps15, and Atg14L and generates the omegasomes-PI(3)P-enriched subdomains of the ER, where autophagosome formation is initiated [37,41].

PI3KC3 I complex I is composed of Vps34, Beclin-1, Vps15, and UVRAG, instead of Atg14L [42,43]. Whereas Atg14L collects mainly in the autophagosome, UVRAG localizes to early and late endosomes [42,44]. The PI3K complex I also participates in autophagosome formation and depends on the interaction of UVRAG with Beclin1 [45]. Interestingly, UVRAG also engages Vps34 to the autophagosome and stimulates Rab7 promoting the fusion of autophagosomes with late endosomes/lysosomes. The activity of UVRAG depends on the v-ATPase [44].

Proteins, such as WIPI2B (WD repeat domain phosphoinositide interacting 2b) and DFCP1 (Double FYVE Containing Protein) that contain PI3P-binding domains, are recruited and promote the expansion of the isolation membranes IMs [46,47]. WIPI2 binds Atg16L1 (Atg 16 like 1) and recruit another complex composed of Atg12, Atg5, and Atg16L1 that stimulates the conjugation of Atg8 family proteins-which comprises LC3 (microtubule- associated protein light chain 3) proteins and GABARAPs (γ-aminobutyric acid receptor-associated proteins) to membrane PE (phosphatidylethanolamine). This event gives rise to LC3II-the lipidated form of LC3, a signature of the autophagic process. Atg8s interacts with members of the autophagic machinery containing the LC3-interacting region (LIR), including LIR-containing cargo receptors, and ultimately regulates the elongation and closure of the autophagosomes carrying the material to be degraded. The membranes required for the elongation of autophasomes are recycled from various cell organelles, such as mitochondria, Golgi, and recycling endosomes [33,48]. A deterioration of the energy level also triggers autophagy, which is sensed by AMPK through the ratio of ATP/AMP. AMPK activates TSC2, which in turn inhibits mTOR [49]. Beyond starvation and decline of cell energy, autophagy can be induced during development and by stressors, such as hypoxia and infections, allowing the disposal of unwanted material, such as aggregated proteins, damaged organelles, and invading pathogens. 

Meanwhile, the dephosphorylated form of TFEB translocates to the nucleus and activates the transcription of autophagy-related genes and genes of the CLEAR (Coordinated Lysosomal Expression and Regulation) network that ultimately control the lysosomal biogenesis [50,51]. In this situation, the levels of Arl8 and motor protein KIF2 (Kinesin-related protein 2) diminish, and both lysosomes and autophagosomes move towards the nucleus where they fuse, leading to degradation of autophagosome contents to alleviate starvation, maintaining the cell alive [29]. Arl8 also promotes the fusion of lysosomes with autophagosomes and endosomes [52]. Fusion of autophagosomes/amphisomes with late endosomes/lysosomes depends on a protein complex formed by the Qa SNARE STX17 (sintaxin 17), the Qbc SNARE SNAP29 (Synaptosome Associated Protein 29), and the R-SNARE VAMP8 (Vesicle-Associated Membrane Protein 8) to lysosomes [53]. HOPS interact with SYX17 mediating the assembly of SNAREs [54,55], and this interaction of HOPS and STX17 is essential for HOPS to tether autophagosome to lysosome ensuring their fusion.

An autophagosome fuses to numerous lysosomes so that autophagy provokes a depletion of the latter. To prevent this, besides controlling the biogenesis of new lysosomes, autophagy also triggers an alternative mechanism of lysosomal biogenesis, named the autophagic lysosomal reformation mechanism that consists of restoring them from tubular structures derived from autophagosomes [56].

In conclusion, lysosomes sense and respond to many stimuli that reflect regulated degradation requirements, such as starvation, accumulation of defective proteins, membrane repair, pathogen infection, etc. These kinds of stimuli may overwhelm the endolysosomal system depending on their intensity, and lysosome functioning can be drastically altered. It was recently proposed that lysosomes display a set of signs that characterize a lysosomal stress response, which would be (1) increased intralysosomal pH, (2) increased lysosome size, (3) membrane permeabilization, (4) cationic eflux, (5) repositioning intracellularly, (6) misfolded protein aggregation, (7) LDL cholesterol accumulation, (8) redox catastrophe, and (9) bioenergetic crisis [57].

Concerning increased intralysosomal pH, although it was proposed that perinuclear lysosomes are more acidic than peripheral ones (where a higher permeability to protons and a less active V-type ATPase would lead to an increased pH) [58], a recent study using a genetically encoded fluorescence-based probe demonstrated that peripheral lysosomes are as acidic as perinuclear ones [59]. So some issues still need to be resolved. Concerning cellular location, the lysosomal position is dynamic and also associated with its different functions [2]; for instance, lysosome positioning close to the plasma membrane can help enhance mTORC1 activation because activators such as Akt1 are close by.

## 2. How Do Coronaviruses Subvert the Endosomal System

The lysosomal system is crucial for the cell response against pathogens. It may address endocytosed pathogens for degradation, trigger autophagic responses to isolate and degrade microorganisms, detect pathogen antigens that stimulate innate immune responses, process pathogen molecules for antigen presentation. On the other side, pathogens developed many strategies to subvert the endosomal system and overcome this hostile environment. We will now explore the interplay between coronaviruses, and SARS-CoV-2 in particular, with this system and how it affects the infection outcome.

### 2.1. Virus Entry Mechanisms

Viruses come in different shapes and sizes; they can be naked or enveloped, they have different nucleic acid content hence distinct strategies of replication and, finally, specific immune evasion strategies. Despite these differences, they all share a viral cycle usually comprised of six phases: (1) host cell binding, (2) host cell entry, (3) uncoating, (4) replication, (5) assembly, and (6) release from the cell. Here, we intend to show the latest findings on the role of the endolysosome system in some of these phases during SARS-CoV-2 infection. 

The first hurdle a virus must overcome is how to enter cells protected by a lipid membrane. Viruses first have to attach themselves to the membrane using host receptors, and then either they fuse their envelope to the plasma membrane or induce their endocytosis. Few viruses merge directly into the plasma membrane, transferring their capsids to the cell cytoplasm [60]. Most viruses invade cells high-jacking the endocytic machinery, which provides many benefits. As no viral proteins are left at the plasma membrane, immune surveillance fails to recognize infected cells. Besides, since the virus is inside a vesicle, it avoids both the microfilament meshwork and cytoplasm crowding and also the cytoplasmic molecular systems of viral detection, such as RIG-I (Retinoic acid-Inducible Gene I), MDA5 (melanoma differentiation-associated protein 5) and NLRP (Nucleotide-binding oligomerization domain, Leucine rich Repeat and Pyrin domain containing). On the other hand, endosomes do have molecular mechanisms to detect invaders, such as IFITM (Interferon (IFN)-Inducible TransMembrane), TLR (Toll-Like Receptor)3, TLR7, TLR9, which the virus have to deal with [61]. 

When using plasma membrane as an entry site, lipid rafts are the leading choice for many viruses. These micro-domains are rich in sphingolipids and cholesterol, contain several GPI (glycosylphosphatidylinositol)-anchored proteins in the outer leaflet, acylated proteins in the cytosolic leaflet, and transmembrane proteins with long hydrophobic transmembrane domains. These domains are not exclusive of the plasma membrane and can also be found in endomembranes and extracellular vesicles. Due to its characteristics, lipid rafts concentrate several signaling molecules and receptors that viruses can exploit [62].

Caveolae are a sub-type of lipid rafts abundant in proteins of the caveolin family that form membrane invaginations of 50–100 nm, whose formation depends on Caveolin-1 (or on Caveolin-3 in the case of skeletal muscle cells). Caveolae fission into vesicles by the action of dynamin, which is not required for other raft-dependent endocytosis. Mechanisms that do not involve dynamin or clathrin rely on actin polymerization, allowing the vesicle scission by a mechanical mechanism [63].

Caveolae are the route of choice for some viruses, the best-known example being SV40 virus (simian vacuolating virus 40 or simian virus 40)-a nonenveloped DNA virus-that binds to the caveolar ganglioside GM1 (monosialotetrahexosylganglioside) [64,65,66]. After entering, it reaches early and late endosomes, from where it accesses the endoplasmic reticulum, before going to the cytoplasm or directly to the nucleoplasm, where it replicates. Interestingly, caveolae do not seem to accommodate a general endocytic function since they are relatively static, do not have many identified cargoes, and their absence does not compromise the viability of caveolin-1-deficient mice [67,68]. Furthermore, studies performed in the last decade indicated that caveolin-1 inhibited endocytosis [69,70,71] and that virus internalization via caveolae would take up to 12 h compared to raft mediated entry [70,71,72].

A minor population of caveolae, though, moves very rapidly and interacts with other vesicles in a kiss-and-run pattern [73]; therefore, caveolae may represent an alternative pathway for cell entry, which is regulated by caveolin-1, as well as by cholesterol, cavins, and ganglioside [74]. This mechanism is triggered by ligand binding to their receptors located in caveolae and involves specific tyrosine kinases, as elegantly shown by Pelkmans et al. (2005) [73], who explored the differences between clathrin- and caveolae/raft-mediated endocytosis using high-throughput RNAi (RNA interference) approach and automated imaging of SV-40 or VSV infected cells to investigate caveolae- or clathrin-mediated virus entry, respectively. This study unveiled that distinct subsets of kinases are activated during caveolin- or clathrin-mediated endocytosis. The authors found that out of 590 screened kinases, 208 are involved in these entry pathways. Additionally, 92 kinases were specific for VSV infection, 80 for SV-40 infection, and these two entry routes shared 36. Of note, 23 out of these 36 shared kinases enhanced one pathway while inhibiting the other [73]. 

In the early 2000s, studies with SV-40 and GFP (Green Fluorescent Protein) tagged-caveolin-1 mislead researchers to propose that caveolae were internalized in grape-like structures with no molecular markers of the endolysosomal system and a neutral pH, which they termed caveosomes [64]. However, other studies showed that cargos of caveolae traffic to endosomes [75,76], where they can meet cargo from clathrin-mediated endocytosis [77]. Furthermore, SV-40- and cholera toxin-loaded caveolae are directed to early endosomes [78], and caveolin-1 can be detected in early endosomes and recycling endosomes [79,80]. Interestingly, the disassembly of caveolae by downregulation of cavin-1/PTRF (Polymerase I and Transcript Release Factor), another essential component of caveolae, induces the non-caveolar endocytosis of caveolin-1 and accumulation of caveolin-1 in early endosomes and MVB [81]. Nine years after coining the name of caveosomes, the same group claimed that this compartment corresponds to late endosomes, where caveolin-1 accumulates and awaits for degradation after overexpression of exogenous caveolin-1 or disturbance of caveolae assembly [82,83].

Clathrin-mediated endocytosis is a rapid constitutive or ligand-triggered process that provokes the assembly of clathrin underneath the plasma membrane and formation of clathrin-coated vesicles. The fission of these vesicles also depends on dynamin. Cargo passes through early and late endosomes, reaching lysosomes in 30–60 min [84]. Compared to caveolae-mediated endocytosis, it activates a distinct set of protein kinases [73]. Another distinction concerns disassembly of the vesicle coat observed in clathrin-coated vesicles, but not in caveolin-coated vesicles, which are stable. Most viruses invade using clathrin-coated vesicles and fuse around 10–20 min after entering the cell [84]. Caveolae-mediated endocytosis occurs in a slow pace (6 to 12 h after virus internalization) [84]. 

Endocytosis pathways come in various flavors and are far more complicated than the mechanisms described above. Some mechanisms are dependent on RhoA, flotillin, Cdc42, and Arf6 and rely distinctly on dynamin or actin (for a review, see [63,74]). There is also crosstalk between components of distinct endocytic pathways, such as the described inhibition of the non-caveolar clathrin-independent pathway CLIC/GEEC (Clathrin-independent Carriers/GPI-AP enriched endocytic compartments) by caveolar components in a mechanism that does not depend on the formation of caveolae [85]. Moreover, a specific ligand can use distinct routes to enter cells, depending on the membrane domain it is sorted to, with consequences to the cell signaling and fate of the ligand [74,86]. Viruses may also benefit from macropinocytosis induced by growth hormones, apoptotic bodies, and particles. Macropinocytosis is suited for bigger viruses and may also have roles in other steps of the viral cycle, such as egress of the cell [86]. For the reference and details of the mentioned pathways and others, we recommend consulting other reviews, such as one of Mercer and colleagues [84].

After entering the cell, virus-containing vesicles fuse with early endosomes, where viruses localize to the vacuolar domains, being excluded by their size from the narrow sorting tubular domains of endosome. As early endosomes mature to late endosomes, pH drop and activate endosomal hydrolases that cleave viral proteins allowing viral fusion with the endosomal membrane and extrusion of the viral genome into the cytosol. 

We will next describe what is already known about the mechanisms used by SARS-CoV-2 to enter cells.

#### Fusion and Endocytosis Can Take SARS-CoV-2 into Cells

It has been almost two years since the COVID-19 (coronavirus disease of 2019) pandemic emerged. Despite this short time, the scientific community has described its causative agent-the new coronavirus SARS-CoV-2. Scientists drew on previous knowledge of the biology of other coronaviruses, particularly SARS-CoV, and used various models, variants, and pseudoviruses to reveal the basics of SARS-CoV-2, such as how infection and dissemination take place. Therefore, it was possible to develop different vaccine platforms, which have been tested and approved in clinical trials and are proving effective in preventing the development of severe cases of COVID-19 [87].

Coronaviruses are enveloped single-stranded positive-sense RNA viruses that infect various animals, including humans. Human coronaviruses 229E, NL63, OC43, and HKU, are responsible for mild to moderate respiratory illnesses. In contrast, SARS-CoV, MERS-CoV (Middle East Respiratory Syndrome), and SARS-CoV-2, which seem to have originated from spillovers of animal infections, caused the recent epidemics of SARS, MERS, and the current pandemic of COVID-19, respectively. Their viral envelopes have four structural proteins: the envelope (E), the membrane (M), the spike (S), and the nucleocapsid (N) proteins. Here we will dissect the SARS-CoV-2 virus cycle specifically to illustrate how coronaviruses may use or avoid the endolysosomal system to replicate and disseminate.

The spike protein of SARS-CoV-2 and SARS-CoV binds to ACE2 (the angiotensin-converting enzyme 2), the obligate cellular receptor at the plasma membrane [88,89]. Alternative molecules facilitate SARS-CoV-2 infection, such as neuropilin receptors [89], CLR (C-lectin type receptors), CD147 (cluster of differentiation 147) [90], heparan sulfate proteoglycans [91,92], CD-SIGN (dendritic cell-specific intercellular adhesion molecule-3-grabbing non-integrin), and L-SIGN (Liver/lymph node-specific intercellular adhesion molecule-3-grabbing integrin) [93], MGL (Macrophage Galactose-type Lectin), GRP78 (glucose-regulated protein) [94] AXL (AXL Receptor Tyrosine Kinase) [95] and TIM-1 (T-cell immunoglobulin and mucin domain protein 1) [96]. After this step, coronaviruses enter cells by direct fusion with the plasma membrane or by endocytosis and further fusion with the endosomal membrane. Fusion with cell membranes is possible only after proteolytic processing of spike, so the entry site depends on the expression and location of proteases necessary to cleave spike. 

Spike is present in the virus envelope as a trimer. Each monomer of the spike protein has two regions: the S1 head region and the S2 stalk region (Figure 2). The S1/S2 site of SARS-CoV-2 differs from that of SARS-CoV by insertion of four amino acids (RRAR), which consists of the minimal cleavage site for furin. Processing of spike by specific proteases occurs in two steps: priming and activation. The priming step consists of cleavage of spike by furin between these two regions that remain noncovalently associated. The S1 region contains the receptor-binding domain (RBD) that binds the angiotensin-converting enzyme 2 (ACE2), while the S2 is responsible for fusion to cell membranes. [97,98,99]. Furin’s activity also strongly stimulates SARS-CoV-2, and although it is not essential (furin knockout cells are still susceptible to SARS-Cov-2), the virion production is diminished in furin’s absence. Deletion of the multibasic furin site in spike abrogates viral infection, indicating that furin is the main protease involved in priming of spike, but that other unidentified protease may also accomplish this task (Figure 2A) [100].

The RBD has a dynamic conformation that may exhibit two states: the “up” and the “down” positions, the first allows binding to the ACE2 receptor but exposes the virion to immune surveillance, and in the “down” state, binding to ACE is hindered. Although SARS-CoV-2 RBD has a higher affinity for ACE2 than SARS-CoV RBD, it is usually held in the “down” state, which may explain why SARS-CoV-2 evades immune surveillance more effectively than SARS-CoV [101,102,103]. The acquisition of immune evasion features is commonly associated with diminished fitness and infectivity. Nevertheless, SARS-CoV-2 is highly infective. One explanation might come from the presence of the RRAR polybasic motif in the SARS-CoV-2 spike that allows preactivation by the cellular protease furin. Whereas the priming step occurs at the plasma membrane during SARS-CoV entry, it seems to occur during cell exit of SARS-CoV-2 virions so that their spike proteins are already primed when these virions reach neighboring cells [97,101]. In other words, SARS-CoV-2 acquired an advantageous modification compared to SARS-CoV, which needs cathepsins activity in the endosomal environment to infect cells successfully. 

A second cleavage site (S2′) upstream to the S2 site allows activation. When cells express TMPRSS2 (the transmembrane serine protease), SARS-CoV-2 fuse directly with the plasma membrane. The activity of this enzyme exposes hidden hydrophobic residues of the S2 region that rapidly interacts with the host cell membrane activating fusion of the viral and host membranes within 10 min and ejection of the viral genome to the cytoplasm. In TMPRSS2-low expressing cells, virus entry occurs by endocytosis and can take up to one hour [104]. In this case, when the virus reaches the endosomes, it encounters cathepsins (non-specific low pH activated proteases) that activate both SARS-CoV and SARS-CoV 2 spike proteins [99,105,106,107,108] (Figure 2B). 

MERS-CoV also exhibits a multibasic motif in its S1/S2 site and likewise enters cells after sequential processing by furin at the S1/S2 site and TMPRSS2 at the S2′ site [97,109]. Genetically-engineered MERS-CoV pseudovirus displaying an uncleavable S1/S2 motif efficiently infects Huh.7 and Vero81 cells, but not the human lung cell Calu-3 nor the human airway epithelial (HAE) cells. The administration of camostat-an inhibitor of serine proteases including TMPRSS2-or E64d-an inhibitor of cathepsins-proved the differential protease usage dependent on the cell type. Indeed, qRT-PCR revealed that Huh.7 contains furin, cathepsins L and B, but few TMPRSS2 transcripts compared to Calu-3 cells, whereas this latter cell type expresses more TMPRSS2 and less furin and cathepsins L and B transcripts. Transduction with cathepsin genes sensitizes Calu-3 to uncleaved MERS-CoV pseudovirus [110].

It is not surprising that TMPRSS2 is expressed in the surface of SARS-CoV-2 main targets: respiratory, gastrointestinal, and urogenital epithelia [98]. Accordingly, efficient in vivo infection requires TMPRSS2 activity. Using rodent models of infection, Zhou and colleagues showed that camostat protects mice from SARS-CoV-induced mortality, and Iwata-Yoshikawa and colleagues showed that SARS-CoV- or MERS-CoV-infected TMPRSS2-deficient mice exhibit reduced viral replication mirrored by a less severe lung pathology and inflammatory response [111,112]. 

The replication of SARS-CoV occurs relatively later compared to other viruses that escape late endosomes and reach the cytosol. Moreover, SARS-CoV traffics deep and invade the cytosol only after entering endolysosomes. It occurs that endolysosomes show the optimal pH for cathepsin L activity, compared to late endosomes and early endosomes that exhibit low or no cathepsin L activity, respectively. Importantly, besides being critical for spike activation, endosomal proteases may also inactivate it due to extensive proteolytic activity depending on how long spike is exposed to them, so the time viruses spend within the endocytic compartment is crucial for a successful infection [113,114,115]. Viruses envelope fuses to the endosomal membrane through a similar process described for the plasma membrane, ejecting the viral genome into the cytosol.

HEK293T cells express shallow levels of ACE2 or TMPRSS2 compared to other cells [116]; hence exogenous expression is usually performed to assure SARS-CoV-2 infection. Bayati and coworkers using HEK293T-ACE2^hi^ and a spike-pseudotyped lentivirus demonstrated that infection occurred by clathrin-dependent endocytosis, inhibited by a clathrin heavy chain knockdown approach. [117]. Moreover, the endocytosis of purified SARS-CoV-2 spike protein is also inhibited after clathrin knockdown or after treatment with either dynasore that blocks dynamin or Pitstop 2 that impairs clathrin-coated pit formation [117]. On the other hand, Li and coworkers used a retrovirus pseudotyped SARS-CoV-2 spike protein to show that these particles enter HEK293T-ACE2^hi^ cells by clathrin- and caveolin-independent processes. These conclusions were evidenced by the lack of inhibition on the entrance of pseudotyped SARS-CoV-2 by chlorpromazine (inhibits clathrin-mediated endocytosis by an unknown mechanism) and dynasore, as well as by the silencing of clathrin, caveolin, EA2, and dynamin [118]. However, the entry process is sensitive to lysosomotropic agents that mediate acidification and methyl-beta-cyclodextrin (MβCD)-that depletes cholesterol, indicating that pseudoviruses possibly use raft-dependent endocytotic mechanism [118]. Results from our group confirm these findings on the role of cholesterol in SARS-CoV-2 entry using the human epithelial lung Calu-3 cells model. These cells express ACE2 and TMPRSS2, allowing SARS-CoV-2 infection by fusion with the plasma membrane [97,99]. We observed that simvastatin-an inhibitor of cholesterol synthesis-inhibits the adsorption and internalization of SARS-CoV-2 into Calu-3. This drug also provokes displacement of ACE2 from rafts diminishing SARS-CoV-2 infection in Calu-3 cells. As entry into these cells occurs preferentially through the fusion of viral envelope with the plasma membrane, these results indicate that cholesterol may be necessary for proper compartmentalization of ACE2 in rafts, but not necessarily because of raft-mediated endocytosis (Teixeira et al., submitted). In fact, rafts are also implicated in viral fusion with cell membranes [62]. This may also be true for membrane fusion with endosomal membranes, since cholesterol depletion inhibits the entry of SARS-CoV in Vero.E6, which occurs through endocytosis. In this study, ACE2 was not found to localize in rafts [119]. 

The role of cholesterol in coronaviruses infection and syncytia formation was specifically addressed in several papers. Treatment with 25HC (25-cholesterol) provokes the depletion of accessible cholesterol from the plasma and inhibits the fusion of coronaviruses with the plasma membrane, inhibiting the formation of syncytia and infection of SARS-CoV, MERS-CoV, and SARS-CoV-2 [120,121]. Mechanistically, Wang and colleagues demonstrated that 25HC induces the activation of acyl-CoA:cholesterol acyltransferase (ACAT) that esterifies accessible cholesterol from the plasma membrane, which is then stored in lipid droplets. They showed inhibition of infection in Calu-3, Caco-2 cells, and lung organoids [120]. Zang and coworkers showed that 25HC accumulates in late endosomes and that dominant-negative mutants of Rab5 and Rab7 diminish pseudovirus SARS-CoV-2 infection in a non-additive manner with the 25HC treatment. In this study, HEK293-ACE2+TMPRSS2- cells were used, implying that results reflect the effects of 25HC in a model that employs the endosomal entry pathway. The authors excluded possible effects of 25HC on ACE2 surface levels, S cleavage by TMPRSS2, lipid raft localization, plasma membrane fluidity, endosomal pH, and its ability to bind to recombinant SARS-CoV-2 spike protein [120] directly. Notably, 25HC is synthesized by CH25H (cholesterol-25-hydrolase), which is coded by an ISG (interferon-stimulated gene) after SARS-CoV-2 infection in vitro and in COVID-19 patients [121]. 

Interestingly, Zang and colleagues observed that 25HC affects pseudovirus entry only after pretreating the pseudovirus, but not the host cell with the drug. On the contrary, Sanders and colleagues observed that cholesterol is required for SARS-CoV-2 entry on the virus membrane, instead of the ACE2+/TMPRSS2+ A549 cells plasma membrane [122]. The authors showed that cholesterol is also required for ACE2+ cell-Spike+ cell fusion and that fusion greatly relies on the rich cysteine-content of coronaviruses spike, which is especially high in SARS-CoV-2. Cysteine residues are important for the covalent binding of fatty acids to proteins. Consistently, 2-BP (2-bromopalmitate)-which inhibits palmitoylation–diminishes cell-cell fusion in U2OS (human sarcoma) and Vero cells. Of note, membrane cholesterol required for fusion does not seem to reside in rafts. Myriocyn, which disrupts rafts through depletion of sphingomyelin of these domains, did not inhibit fusion. Another critical point of this study is the drugs’ distinct effects depending on the model used. 25HC, MβCD, and zaragozic acid-which inhibits cholesterol synthesis–inhibit syncytia formation of ACE2+-U2OS cells. The entry of Spike-pseudotyped MLV (mouse leukemia virus) in ACE+/TMPRSS2+ A549 cells was not inhibited by 25-HC, contrary to the studies described above. However, it was hindered by MβCD and Apilimod, which inhibits PIKfyve-the main enzyme-synthesizing PI(3,5)P2 (phosphatidylinositol 3,5-bisphosphate)-also responsible for maturation of early to late endosomes. Notwithstanding, MβCD, which was shown to abrogate entry of pseudovirus and cell-cell fusion, did not inhibit the infection of ACE+/TMPRSS2+ A549 cells by a clinical isolate of SARS-CoV-2 [122]. It must be noted that cholesterol may have a wide range of effects beyond its role in the composition of the plasma membrane.

The participation of PYKfyve in SARS-CoV-2 and SARS-CoV entry was also shown by the use of Apilimod and Vacuolin-1 by Kang and colleagues [123], and the corresponding gene was identified in a genome-wide screening for host factors [124]. Inhibition of TPC2, but not of TRPML1-two major downstream effectors of PI(3,5)P2-also impair SARS-CoV-2 invasion [107]. 

The entry of the SARS-CoV virus is also controversial, probably reflecting the particularities of each of the models used in the studies. SARS-CoV spike protein-bearing pseudotypes invade HEK293 and Vero.E6 cells (cells that do not express TMPRSS2) by a clathrin-and caveolin-independent mechanism, which is sensitive to alkalization by lysosomotropic agents and cholesterol depletion by MβCD [125]. Nevertheless, entry of an HIV-based pseudovirus bearing the SARS-CoV spike protein into HepG2-cells that do not express caveolin-1-is sensitive to chlorpromazine and clathrin heavy chain silencing, implying the involvement of clathrin in this process [126].

Importantly, SARS-CoV-2, as other enveloped viruses, can invade neighboring cells through induction of direct fusion of infected- and non-infected cells, thus forming multinucleated giant cells or syncytia. This is possible due to fusogenic proteins left in the cells’ surface after virus fusion with the plasma membrane. The spike protein of SARS-CoV-2 infected cells interacts with ACE2 of neighboring cells, promoting fusion [127] (Buchrieser et al., 2020). Whereas SARS-CoV-2 and MERS-CoV spike proteins are highly fusogenic, SARS-CoV spike is not, concurrent to the requirement of the polybasic motif that is present only in SARS-CoV-2 and MERS-CoV for fusion occurrence. Trypsin treatment or TMPRSS2 expression increases the formation of syncytia [97]. As for virus fusion with the plasma membrane, furin cleavage is not essential but increases cell-cell fusion [97,100].

For an extensive review of the molecular and cellular events involved in SARS-CoV-2 entry into cells, we recommend reading Jackson and col. [128].

### 2.2. Replication and Assembly of Coronaviruses

Once the virus fuses to the endosomal or plasma membrane, its positive-sense single RNA genome is ejected to the cytosol, where it is translated. The coronavirus genome comprises 14 open reading frames (ORFs), organized in genes coding the replicase, structural and accessory proteins. The replicase gene comprises two large open reading frames, ORF1a and ORF1b. Viral proteases encoded by the ORF1a cleave the two large viral polyproteins translated from ORF1a and ORF1ab, giving rise to 16 nonstructural proteins (NSPs) involved in the viral genome replication [129]. Meanwhile, cell endomembranes are remodeled into replication organelles (ROs), composed of double-membrane vesicles (DMV), which protect the viral RNA from cytosolic sensors of dsRNA that are part of the innate immunity response. Full length and numerous sub-genomic viral RNAs leave DMVs and reach the cytosol through membrane pores that span both membranes of DMVs and are mainly constituted by NSP3 and probably by other components of viral and host origin [130]. The small RNAs are translated into viral structural and accessory proteins. The N protein is cytosolic and binds the full-length viral RNA condensing it. The N protein also interacts with NSP3, which may help guide the viral RNA to the replication sites and encapsulate the viral RNA when it leaves DMVs and reaches the cytosol [129]. The structural M, S, and E proteins are transmembrane proteins that traffic from the ER to the Golgi for glycosylation and then go to the intermediate compartment between the endoplasmic reticulum and the Golgi apparatus (ERGIC). Viral assembly occurs when N-coated viral RNA buds into the lumen of ERGIC forming new virions, whose envelope is now covered with the structural proteins S, E, and M [131,132,133]. It was believed then that coronaviruses left the cells through the biosynthetic secretory pathway, but Ghosh and collaborators challenged this view, elegantly demonstrating that mouse hepatitis virus (MHV) and SARS-CoV-2 rather use the lysosomal exocytotic pathway [134] (Ghosch et al., 2020). For a more detailed view of the coronavirus cycle, we recommend the reviews of V’kovski and Bracquemond [135,136].

SARS-CoV2 still has one obstacle to overcome, which is to get out of the cell to continue spreading. We will now look into ways SARS-CoV2 emerges from infected cells.

### 2.3. Exit from the Cell: Lysosomes as Alternative Routes

It was in the ’80s that coronaviruses were reported to be localized in lysosomes with no sign of being destroyed in these organelles [137]. Currently, there is a large body of evidence showing a vital role of lysosomes in the repertoire of methods coronaviruses use to ensure infection.

Ghosh and collaborators used the in vitro models of HeLa-mCC1 cells and primary macrophages infected with MHV and Vero.E6 cells infected with SARS-CoV-2 to study the mechanisms coronaviruses engage in exiting infected cells. At 6 h post-infection, coronaviruses mainly replicate and assemble the new virus particles, and the cell begins to release virions to the extracellular medium. Trafficking through Golgi and TGN is essential for assembly, evidenced by colocalization of viruses with TGN46, Golgin97, and mannosidase II at 6h. However, at 12 h post-infection, viruses are no longer in these organelles when viral egress reaches its peak. Still, they colocalize within LAMP1+ and Cathepsin D+ vesicles, which correspond to late endosomes and lysosomes. Accordingly, blocking anterograde biosynthetic traffic of secretory vesicles using brefeldin A 6 h after infection did not affect viral egress [134].

Notably, the authors ruled out the possibility of colocalization of viruses and endolysosomes/lysosomes due to the reuptake of released virus by blocking endocytosis with Dyngo-4a, a potent dynamin inhibitor. Inhibitors of dynamin-independent endocytosis were not used, though, which could be explored in the future to exclude this possibility. Lysosomal exocytosis was evidenced by the increase of LAMP1 labeling at the plasma membrane and the release of pro-Cathepsin D and matured Cathepsin D at the extracellular medium [134].

Arl8b, implicated in the anterograde motility of lysosomes, colocalizes with LAMP1+ and MHV+ vesicles, and the corresponding gene silencing inhibits MHV infection. Moreover, the authors show that viral release is not a consequence of the fusion of late endosomes/MVB with the plasma membrane through depletion of Rab27 (implicated in MVB anterograde motility and fusion to the plasma membrane) or use of GW4869 (an inhibitor of ESCRT-independent biogenesis of exosomes). On the other hand, the Rab7-selective competitive nucleotide-binding inhibitor CID1067700 prevents MHV egress, an effect associated with a decrease of LAMP1+ organelles, corroborating the importance of lysosomes to this event. Significantly, infection with MHV and SARS-CoV-2 reduced the acidity from pH 4.7 to pH 5.7, increased secretion of pro-Cathepsin D, and reduced protease activity. More than an alternative way to exit cells, an important outcome of this viral strategy is the reduced cell ability to process and present antigens [134], which helps viral escape from immune surveillance (Figure 3).

Lysosomes of SARS-CoV-2 infected cells are positive for the open reading frame 3a (ORF3a). We will refer to SARS-CoV-2 ORF3a as ORF3 hereafter unless the viral origin needs to be specified. Interestingly, when this protein is ectopically expressed in Vero.E6 cells, acidification of lysosomes is hampered compared to ORF3a non-expressing cells, indicating that ORF3a protein may be related to deacidification of lysosomes [134]. SARS-CoV ORF3a ectopically expressed in HeLa cells localizes to LAMP1+ lysosomes and provokes the release of cathepsins to the cytoplasm, impairing lysosomal degradation capacity. Furthermore, SARS-CoV ORF3a induces TFEB translocation to the nucleus and an increase in p62 and LAMP1 levels of expression [138]. These data point to the role of lysosomes as a signaling hub for controlling the cell response to coronavirus infection. Indeed, the literature reveals the central role that SARS-CoV-2 ORF3a plays during viral egress modulating lysosome function to benefit the virus and success of viral infection [139,140,141,142].

Miao and coworkers performed a series of elegant experiments that show ORF3a can block the autophagic flux in cells. ORF3a is a transmembrane protein and, when expressed in HeLa cells, localizes to Rab7- and LAMP1-+ late endosomes/lysosomes, but not to EEA1-+ early endosomes and LAMP2-+ lysosomes. In SARS-CoV-2 infected cells and ORF3a-expressing cells, late endosomes/lysosomes accumulate, whereas the number of EEA1+ early endosomes does not change. Furthermore, ORF3a interacts with Vps39 and Vps41, the specific sub-units of the HOPS complex, which is involved in tethering late endosomes/lysosomes to amphisomes and autophagosomes [140].

On the other hand, ORF3a impairs the interaction of endosomal/lysosomal HOPS with amphisomal/autophagosomal STX17 and the formation of the STX-17/SNAP29/VAMP8 SNARE complex. Thus, the authors suggest that the viral ORF3a protein localized in late endosomal/lysosomal sequestrates the HOPS complex component Vps39, preventing HOPS from interacting with the autophagosomal SNARE protein STX17. This blocks assembly of the trans-SNARE complex composed of STX17, SNAP29, and VAMP8, which drives the fusion of autophagosomes and amphisomes with lysosomes [140] (Figure 3).

Significantly, ORF3a expression also affects lysosomal biogenesis, increasing Rab7-, LAMP1- and Lysotracker-labeled late endosomes/lysosomes, but not of EEA1-labeled early endosomes. Nevertheless, even though ORF3a induces nuclear translocation of TFEB-which regulates autophagy and lysosomal biogenesis-most of the analyzed genes involved in lysosomal biogenesis and autophagy were not altered. Furthermore, ORF3a provokes lysosomal damage, characterized by dysfunctional lysosomal proteases and exposure of Galectin-3-labeled luminal lectins [140]. 

Remarkably, SARS-CoV ORF3 also disturbs lysosome function, increasing p62 expression and TFEB nuclear translocation [138]. However, it does not interact with Vps39 nor affects autophagosome maturation [140]. 

Other studies have also demonstrated ORF3a blocking the fusion of the autophagosome with lysosomes. Zhang and colleagues described a very similar mechanism depicted by Chen and colleagues involving the ability of ORF3a to bind Vps39 and subsequent inhibition of the interaction of HOPS with the fusion machinery. Specifically, they showed that ORF3a impedes binding of HOPS and Rab7 [142], which is required for amphisome/autophagosome maturation [47]. Qu and colleagues identified another mechanism behind the ORF3a effect on autophagy [141]. In this study, the authors show that ORF3a interacts with UVRAG, favoring the formation of the PI3KC3 complex I (Beclin-Vps34-Atg14) and inhibiting the formation of the PI3KC3 complex II (Beclin-Vps34-UVRAG). As in the former cited studies, the SARS-CoV ORF3a does not show such an effect [141]. Taken together, these results suggest that ORF3a can block the autophagic flux in cells.

In all of these studies, all proteins of SARS-CoV-2 were screened for their potential to modulate autophagy. Although other proteins, such as ORF7a, M, Nsp5, Nsp6, and Nsp8 also seem to have an effect, ORF3a exhibited the most potent inhibitory activity [140,141,142]. Accordingly, the Nsp6 of MHV, SARS-CoV, and the avian coronavirus IBV (infectious bronchitis virus) block the autophagic flux at the level of autophagosome expansion [143].

The interaction of ORF3a with Vps39, already demonstrated by Miao and colleagues, [140] proved to be important for blocking autophagy flux and increasing lysosomal exocytosis. Indeed, knocking down Vps39 inhibits the formation of BORCS6, VAMP7, and STX4 complex and lysosomal exocytosis, evidenced by a drastic reduction of LAMP1 in the plasma membrane [139] (Figure 3).

Of note, the infection with SARS-CoV2 induces a drastic increase in the formation of VAMP7 and STX4 puncta, likewise. Interestingly, the coronavirus MHV uses lysosomal exocytosis to exit the infected cell but does not have an ORF3a in its genome. This suggests that another viral protein or even another mechanism is involved in this process. Notwithstanding, SARS-CoV-2 ORF3a expression in MHV-infected cells increases the titer of released MHV, indicating that SARS-CoV-2 ORF3a potentiates MHV infection, possibly by facilitating MHV egress [139]. 

SARS-CoV ORF3a, which lacks two residues present in SARS-CoV-2 ORF3a-serine 171 and tryptophan 193, cannot interact with Vps39 and promote the STX4 and VAMP7 complexes [139]. When SARS-CoV-2 ORF3a is mutated in one of these sites, the effects mediated by ORF3a are reduced. Not surprisingly, when these residues are introduced at equivalent positions in SARS-CoV ORF3a, they provide this protein the capacity to promote both lysosomal exocytosis and autophagy [139]. Nevertheless, SARS-CoV ORF3a has other functions and can provoke apoptosis [144], NLPR3 inflammasome activation [145], and chemokine production [146], and its deletion is protective in a mouse model of infection [147].

The relevance of autophagy in SARS-CoV-2 infection was further demonstrated in a hamster model of SARS-CoV-2 infection and in samples of COVID-19 patients [148]. The authors used a very low MOI (multiplicity of infection) of SARS-CoV-2 to infect VeroFM (IFN-deficient cells) and Calu-3 cells in this study. A high ATP/AMP ratio is preserved in these models, and there is amino acid sufficiency. Coherently, there is a reduction of AMPK activity, while mTORC1 is kept active and the phosphorylation of ULK1 is close to the basal level during the time course of infection. The authors suggest that SARS-CoV-2 infection may inhibit host translation in their models, avoiding starvation-triggered autophagy and thus maintaining the cellular energy and nutrient status in balance [148].

Nevertheless, SARS-CoV-2 reduces the autophagy flux, evidenced by the activation of the autophagy inhibitors AKT1 (Akt serine/threonine kinase 1) and SKP2 (S-phase kinase-associated protein-2) and the decrease of proteins involved in autophagy initiation (AMPK, TSC2, and ULK1), in membrane nucleation, phagophore formation, and autophagosome/lysosome fusion (Beclin1, VPS34, ATG14). Furthermore, there is a reduction of fusion of the autophagosome with lysosomes, concomitant with an increase of LC3-II/actin ratio. To investigate the impact of SARS-CoV-2 infection on the autophagic flux in vivo, Syrian hamsters were infected and analyzed at different time points. Lung samples of SARS-CoV-2-infected hamsters exhibit an increase of p62 and LC3-II compared to control, which is notably higher in aged hamsters. This effect was attributed to the inhibition of autophagy because the mRNA levels of MAP1lc3b and Sqstm1, coding for LC3 and p62, respectively, remained unaltered. Finally, the occurrence of p62 and LC3-II was higher in lung sections of COVID-19 patients compared with lung sections of pneumonia patients or deceased by other causes. Of note, the genes involved in autophagy are differentially regulated depending on the cell type and disease duration, and SARS-CoV-2 replication levels. Notably, the use of drugs and metabolites that affect autophagy proved helpful in the control of SARS-CoV-2 infection. Spermidine and spermine inhibit the production of infectious particles, in agreement with the elevated levels of putrescine found after SARS-CoV-2 infection. Rapamycin–an inhibitor of mTOR, induces autophagy and diminishes the production of infectious particles. Inhibitors of ULK1 (MRT6821) and Vps34 (SAR405) stimulate SARS-CoV-2 infection. Knockdown of players in the autophagic process, such as ATG5, ATG7, and FIP200 promote SARS-CoV-2 infection. Of particular interest, MK-2206–an AKT inhibitor in clinical phase II trial-induces autophagy by upregulating Beclin1 and reduces the production of infectious SARS-CoV-2 by up to 92%. Accordingly, knockdown of Beclin1 increases SARS-CoV-2 replication and the use of SMIP004, valinomycin, and niclosamide–inhibitors of SKP2, which block Beclin1 induction of autophagy–suppress the production of SARS-CoV-2 infectious particles [148].

Liu and coworkers further demonstrated another mechanism of subversion of autophagy by coronaviruses. They demonstrated that infection with the coronaviruses OC43 and 229E promote TFEB degradation by activating the PAK2 kinase that phosphorylates TFEB and primes it for ubiquitin-driven degradation mediated by the E3 ubiquitin ligase subunit DCAF7. The authors identified the agents BC18813 and BC18630 that can interfere with TFEB-DCAF7 interaction. Importantly, these agents attenuated the in vitro infection of Beas-2B cells with OC43 or 229E, MCDK cells with Influenza H1N1, and Calu-3 cells with SARS-CoV-2 while restoring lysosomal function. Finally, BC18630 was also effective in attenuating SARS-CoV-2 infection in vivo using a hamster model [149]. 

Other viral proteins may also affect the host endolysosomal system, such as the S1 region of the spike protein of SARS-CoV-2. Processing the spike protein in the S1 head region and the S2 stalk regions results in the shedding of the S1 spike domain (671 amino acids long) found in various body fluids and tissues. 

Since the S1 spike domain can cross the blood-brain barrier, possible neural effects of S1 were analyzed in neuronal cells, such as the mouse embryonic hippocampal cell line CLU199, mouse embryonic hippocampal neurons, and primary human cortical neurons HNC001. S1 spike domain fused to an N-terminal His and a C-terminal tag was able to enter neurons and accumulate in LysoTracker-endolysosomes and LAMP1+ lysosomes, resulting in deacidification of these organelles and a decrease in the percentage of active cathepsin D lysosomes. It also provoked endolysosomal enlargement and neurite dystrophy, which the authors suggest may be related to the requirement of constant vesicular membrane trafficking. Interestingly the entire SARS-CoV-2 and SARS-CoV spike S1 domain do not elicit the same effects [150]. 

An aspect to consider about using tagged proteins to analyze their fate through the endolysosomal system is the possibility of creating artifacts, such as what is observed in the case of tagged-caveolin-1, which accumulates in late endosomes as described by Hayer and colleagues [82].

## 3. Genome-Wide Studies Reveal Components of the Endolysosomal System as Potential Pharmacological Targets against COVID-19

The advancement in high-throughput technologies allowed the recognition of many genes and cellular signaling networks that are potential targets for therapy against SARS-CoV-2 and COVID-19. 

A study of genome-wide CRISP knockout screening to identify host factors that confer resistance to SARS-CoV-2 and the common cold coronaviruses 229E and OC43, followed by a gene ontology enrichment analysis identified genes essential for this resistance and organized them in biological networks, such as macroautophagy and phospholipid metabolic process, lysosome to endosome transport and membrane fusion, Golgi vesicle transport, Rab protein signaling, phagosome maturation and cholesterol metabolic process. The cells used in this study were Huh7.5.1 for OC43 and 229E infections and Huh7.5.1 cells overexpressing ACE2 and TMPRSS2 to optimize SARS-CoV-2 infection [124]. 

Host genes related to the virus entry were detected, such as the ones coding for ACE2 and ANPEP, the respective receptors for SARS-CoV-2 and 229E, and multiple genes involved in heparan sulfate biosynthesis, involved in infection by OC43 that uses sialic acid or glycosaminoglycans for cell entry. Genes of the cholesterol metabolism pathway were identified in all three coronaviruses screens (SCAP, MBTPS1, MBTPS2, LDLR, and NPC1). The common cold viruses screens exhibit genes related to the phosphatidylinositol metabolism, such as PIK3C3, UVRAG, BECN1, and PIK3R4), involved in endosome sorting, endomembrane homeostasis, and autophagy; some of their products were already mentioned in this review. Specific to these viruses were also the genes involved in the endosome and phagosome maturation (RAB7A, members of the HOPS complex-VPS11, VPS16, VPS18 and VPS33A, the Ccz1-Mon1 guanosine exchange factor complex -CCZ1, CC1B, and C18orf8, the WDR81-WDR91 complex) and others associated to lysosome and autophagosome function (SPNS1, TOLLIP, TMEM41B, AMBRA1). Other genes related to lysosomal enzymes and their traffic from the Golgi to the lysosomes include CTSL1, M6PR, and GNPTAB. 

In the SARS-CoV-2 screen, the top hit was the lysosomal transmembrane protein TMEM106B and VAC14, a member of the PIKfyve complex. This study highlights the central role of the endosomal system for coronaviruses infection. 

Another CRISPR loss-of-function screen of human alveolar basal epithelial carcinoma cells infected with SARS-CoV-2 confirmed the central role cholesterol plays during SARS-CoV-2 infection [151]. These results agree with the cholesterol function during coronaviruses cell entry, as depicted earlier in this review (Teixeira et al., submitted) [118,122,125]. Furthermore, cholesterol may also play an essential part in many other steps of the viral cycle, such as the production of lipid droplets-which fuels infection [152,153], formation of double-membrane vesicles required for the replication organelles, production of membranes for the autophagic process, and virus particle production.

Interactomes are very informative in searching for host factors on which SARS-CoV-2 depends to succeed. Gordon and colleagues expressed 26 out of the 29 SARS-CoV-2 proteins in HEK293/17 cells and concluded that 40% of host interacting proteins relate to vesicles or their traffic [154]. The uncovered interactions (where the name of the viral protein is followed by the bracketed name of the host protein or cellular process) revealed that viral infection may remodel the trafficking of ER and Golgi, modulate ER stress response and modify the vesicle compartments to favor replication or to mediate escape from immune responses: NSP8 (SRP-signal recognition particle), ORF8 (protein quality control in the ER), M (morphology of the ER), NSP13 (organization of the centrosome and Golgi), NSP2 (WASH), NSP6 and M (vacuolar ATPase), NSP7 (Rab proteins), NSP10 (AP2), E (AP3) and ORF3a (HOPS), NSP6 and ORF9c (Sigma receptors implicated in lipid remodelling). These are valuable results since many of these proteins are known drug targets then tested. The authors identified two classes of effective drugs against SARS-CoV-2 infection–protein biogenesis inhibitors (zotatifin, ternatin-4, and PS3061) and ligands of the sigma-1 and sigma-2 receptors (haloperidol, PB28, PD-144418, hydroxychloroquine, clemastine, cloperastine, progesterone and siramesin) [154].

While the studies described above used HEK293/17, Stukalov and coworkers used A549 (lung carcinoma epithelial cells) to express each of SARS-CoV and SARS-CoV2 proteins and performed the interactome of viral and host proteins, as well as the alterations caused in the transcriptome, proteome, phosphoproteome, ubiquitinome of cells expressing the viral proteins [155]. With this approach, the authors recognized shared and specific mechanisms used by these two viruses highlighting the role of ORF3a and ORF8 of SARS-CoV2 in manipulating the autophagy and the TGFβ signaling pathway, respectively. The central role of ORF3a in managing autophagy confirms once more the results reported elsewhere, described in this review [139,140,141,142,148]. ORF3 blocks the autophagic flux, causing the accumulation of autophagy-related proteins (SQSTM1, GABARAPL2, NBR1, CALCOCO2, MAP1LC3A, MAP1LC3B, and TAX1BP1). Based on the interactome results, the authors propose that the ability of ORF3 to abrogate autophagy may be due to the interaction of the ORF3 protein with the HOPS complex (Vps11, Vps16, Vps18, Vps39 and Vps41). The disruptive role of ORF3 and Vps39 was later confirmed by Chen and colleagues’ study [139]. They also suggest the differential phosphorylation of regulatory sites on TSC2, mTORC1 complex, ULK1, RPS6, and SQSTM1, and ubiquitination of critical components, such as MAP1LC3A, GABARAPL2, Vps33A, and VAMP8, may explain the ability of ORF3 to inhibit autophagy [155].

In the same study, but now using Vero cells infected with an MOI of 0.01 of SARS-CoV-2, authors test drug efficacy against the virus. In this case, rapamycin suppressed SARS-CoV-2 replication, but the highest antiviral activities were obtained with the following drugs: gilteritinib-a designated inhibitor of FLT3 and AXL; ipatasertib-an AKT inhibitor; and prinomastat and marimastat-matrix metalloprotease (MMP) inhibitors. Importantly, these drugs exhibited minor effects on cell growth [155].

## 4. Conclusions

In this review, we gathered information about the central role of lysosomes in controlling cellular signaling, sensing cellular stress and the metabolic/energy status, degrading intra- and extracellular materials including cell invaders, communicating with the external environment, and participating in the innate immune response. To perform these varied tasks, lysosomes use many protein complexes, which are mobilized in an orchestrated way, allowing interaction with other vesicles of the endolysosomal system and the plasma membrane. Here, our intention was not to cover in depth these molecular pathways but to present a general view of lysosome biology to help the reader understand these processes and how coronaviruses highjack the endolysosomal system and its functions to their benefit.

Viruses provoke a radical shift in cell metabolism and manipulate the cellular machinery to replicate and disseminate. Endomembranes are remodeled to allow virus internalization, replication, assembly, and egress. Coronaviruses handle the lysosomal apparatus to avoid cellular defense mechanisms such as autophagy, luminal degradation, antigen presentation and ultimately explore these organelles as a route of escape. SARS-CoV-2 exhibits traits that make it especially skillful in exploiting the endosomal system, which is reflected in its high effectiveness to infect and spread. The concentrated efforts of the scientific community to comprehend SARS-CoV-2 unveiled many mechanisms involved in infection at an unprecedented speed. This translates into developing vaccines and promising drugs in under two years. Although this review was certainly born already outdated, it highlights the idea that the best way to win is to know our enemy and its tactics well. All the cited work, and those we fail to mention, help us do that. All the gathered information so far indicates that a promising course of action to control the new coronavirus could be the use of pharmaceuticals that interfere with components of the endolysosomal system. The drugs being tested at present -molnupiravir, which introduces mutations into the viral genome or Paxlovid that inhibits viral protein (Nature 599, 358–359 (2021)) processing act directly against the virus, and we can envisage the emergence of resistant strains. The advantage of developing drugs that would act on components of the endolysosomal system is that one would interfere with virus uptake or egress without selecting the virus.

Summed to that, the use of nanotechnology to optimize drug delivery and efficacy represents a powerful tool against COVID-19 (for a review, see [156]. Niclosamide, an approved anti-helminthic drug that stimulates autophagy, has broad antiviral activity but low oral bioavailability [157,158]. Importantly, it has antiviral activity against Alpha (B.1.1.7), Beta (B.1.351), and Delta (B.1617.2) SARS-CoV-2 variants in human primary lung epithelial cells, as well as in the cell lineages Vero and Caco-2 [158]. Nanotechnology has been used to improve the delivery of niclosamide to treat conditions, such as cancer and now COVID-19 [159,160]. With this purpose, the study of Brunaugh and coworkers report the development of an inhalable formulation of niclosamide associated with lysozyme, which helps the solubilization and delivery to the primary sites of infection of coronaviruses at high dose concentrations and improves the antiviral potency of niclosamide in vitro and in vivo against MERS-CoV and SARS-CoV-2 infection using a murine model [160]. Inhaled and intranasal application of niclosamide was tested in a Phase I trial that revealed that its use is safe, well-tolerated, and presents mild adverse effects, such as transient irritation of the upper airways [161].

While SARS-CoV-2 keeps surprising us with new variants, scientists try to learn and understand its biology better to control its spread around the globe. We hope the knowledge of viral strategies of infection associated with the development of vaccines and the advance of new pharmacological approaches will grant us the end this pandemic soon.

## Figures and Tables

**Figure 1 ijms-23-04576-f001:**
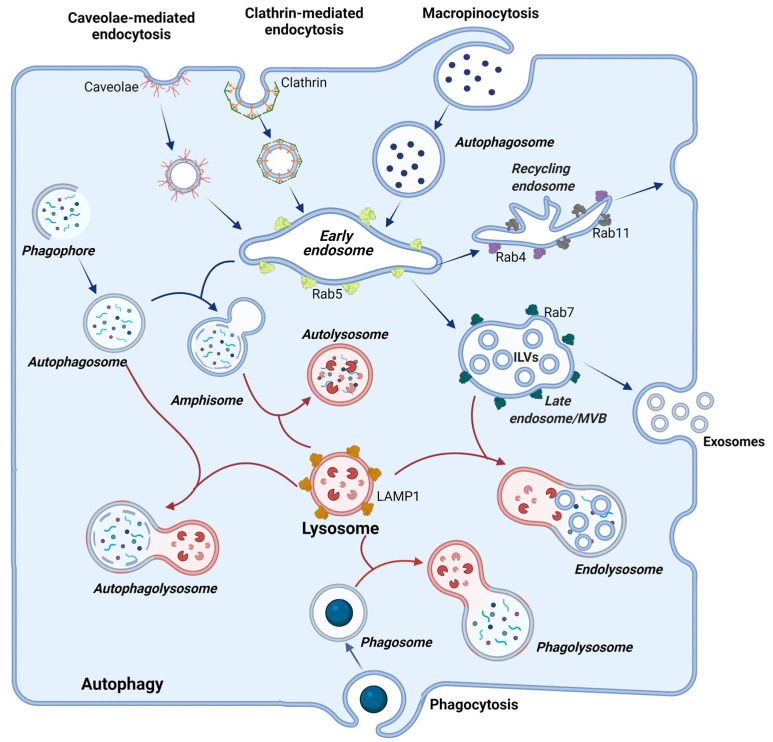
The endolysosomal system. Illustration showing entry pathways into cells and compartments mentioned throughout this review that constitute the endolysosomal system. These vesicles traffic throughout the cell and exchange their contents through ‘’kiss-and-run” or “full-fusion” events. Particles from the extracellular milieu can access cells by caveolae-mediated endocytosis, clathrin-mediated endocytosis, macropinocytosis, and phagocytosis. Particles entering through endocytosis and micropinocytosis are delivered to early endosomes. Vesicles generated during autophagy are depicted on the left side of the figure. Please refer to the main text for a detailed description of the fusion events and specific molecular markers of each of these compartments. Image created using BioRender.

**Figure 2 ijms-23-04576-f002:**
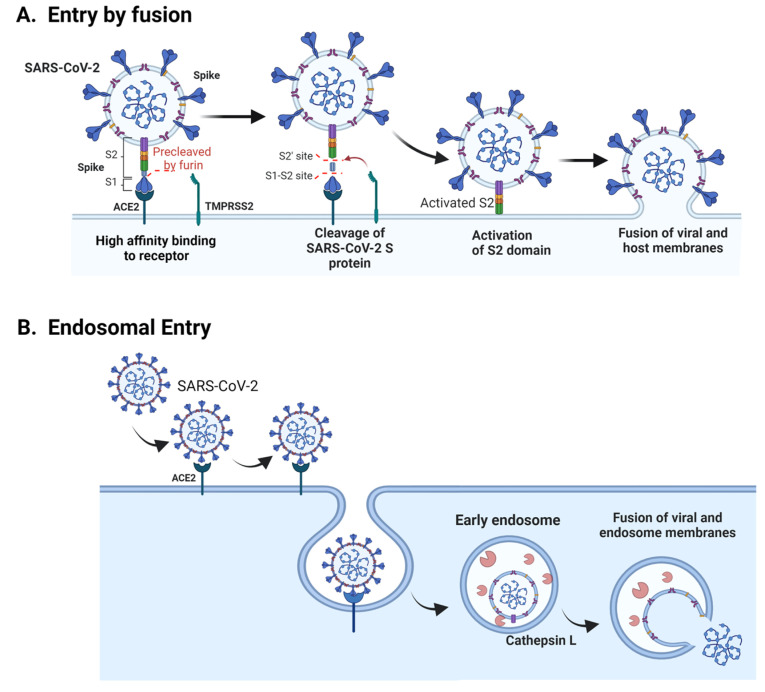
SARS-CoV-2 entry pathways. Illustration depicting ways SARS-CoV-2 uses to enter cells. (**A**) Entry by fusion-the direct fusion of the SARS-CoV-2 envelope with the plasma membrane is mediated by the processing of the spike protein after the interaction with the cellular receptor ACE2. The cleavage sites by specific proteases are shown: furin cleaves spike a multibasic motif in the S1/S2 site (priming step), and TMPRSS2 cleaves spike in the S2′ site (activation step). Cleavage by furin seems to occur during the cell exit of SARS-CoV-2 virions. When cells express TMPRSS2, hidden hydrophobic residues of the S2 region are exposed, and SARS-CoV-2 fuse directly with the plasma membrane. (**B**) Entry by endocytosis-In TMPRSS2-low expressing cells, virus entry occurs by endocytosis. In this case, when the virus reaches the endosomes, it encounters cathepsins (non-specific low pH activated proteases) that activate both SARS-CoV and SARS-CoV-2 spike proteins. Please refer to the main text for a more detailed description of these processes. Image created using BioRender.

**Figure 3 ijms-23-04576-f003:**
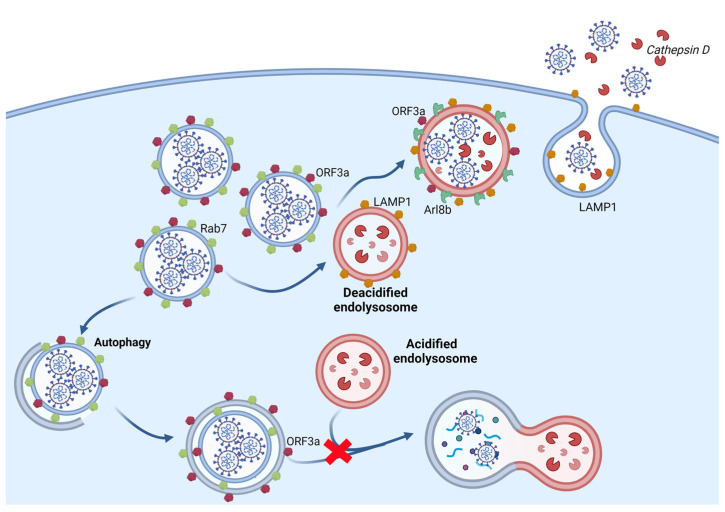
Coronaviruses subvert the endolysosomal system. Coronaviruses infection reduces the acidity of lysosomes from pH 4.7 to pH 5.7 and protease activity. It further provokes lysosomal exocytosis with the release of virions and other contents, such as cathepsin D, to the extracellular environment Lysosomal exocytosis leads to the exposition of lysosomal markers such as LAMP1 at the plasma membrane. The GTPase Arl8b participates in the anterograde motility of LAMP1+ lysosomes filled with viruses. Furthermore, lysosomes of SARS-CoV-2 infected cells are positive for ORF3a, which blocks the autophagic flux in cells. In SARS-CoV-2 infected cells, late endosomes/lysosomes accumulate because ORF3a interacts with the HOPS complex inhibiting the fusion of late endosomes/lysosomes with amphisomes and autophagosomes. For a more detailed description of ORF3a action, please refer to the main text. Image created using BioRender.

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
