# Peer review of "The Endolysosomal System: The Acid Test for SARS-CoV-2"

_ijms, 2022, doi:10.3390/ijms23094576_

Round 1

Reviewer 1 Report

“The endolysosomal system: the acid test for SARS-CoV-2” is a review article. The authors collected an impressive amount of information in this paper regarding the endocytosis of SARS-CoV-2 and the possible action taking place in the endolysosomal system after that. The collected information has great value, and the topic is currently relevant, we still need a better understanding on how the virus can enter the cells. However, the are some issues with the article, which need to be addressed.

The major concern is that the article is hard to read, and it is hard to figure out why some information is important and how they are connecting into the big picture. The authors themselves wrote “our intention was not to cover in depth these molecular pathways but to present a general view of lysosome biology to help the reader understand these processes and how coronaviruses highjack the endolysosomal system and its functions to their benefit”, but they are providing little in terms of explanation and conclusion. A review article needs more structure and should provide (possible) answers to questions the authors want to answer.

It is hard to see how the content in the firs 6 page connects with the second half of the article.

The authors are making comparisons between SARS-CoV and SARS-CoV-2 often, but sometimes it is hard to see the relevance to the topic.

PRRAR sequence can also be a heparin binding domain, and RBD/heparan-sulphate interaction also may have a role in the cell entry mechanism of the virus (10.1016/j.cell.2020.09.033, 10.3390/ijms22105336).

Institutional Review Board Statement: delete text and state “not applicable”.

Author Response

Reviewer 1

 Comments and Suggestions for Authors

“The endolysosomal system: the acid test for SARS-CoV-2” is a review article. The authors collected an impressive amount of information in this paper regarding the endocytosis of SARS-CoV-2 and the possible action taking place in the endolysosomal system after that. The collected information has great value, and the topic is currently relevant, we still need a better understanding on how the virus can enter the cells. However, the are some issues with the article, which need to be addressed.

The major concern is that the article is hard to read, and it is hard to figure out why some information is important and how they are connecting into the big picture. The authors themselves wrote “our intention was not to cover in depth these molecular pathways but to present a general view of lysosome biology to help the reader understand these processes and how coronaviruses highjack the endolysosomal system and its functions to their benefit”, but they are providing little in terms of explanation and conclusion. A review article needs more structure and should provide (possible) answers to questions the authors want to answer.

It is hard to see how the content in the firs 6 page connects with the second half of the article.

Answer: Thank you very much for the criticism about the manuscript structure. Our intention was to first describe some aspects and components of the endolysosomal system for those who do not work in this field. In section 2, we then describe how viruses can subvert this system dividing this section into the following steps of the viral cycle: entry, replication, and exit. Finally, in section 3, we report data generated from genome-wide studies that identify promising candidates for drugs against coronaviruses, some of which participate in the endolysosomal compartment, strengthening the idea that this system is crucial for the viral cycle and could be a target to fight coronaviruses infection. We then come back to some of the components mentioned in sections 1 and 2 as some have been identified in the genome wide studies mentioned above.

Some titles have been modified to make the links among these parts more logical and clear. We hope this leads to a better understanding of the logic we used to write this review.

Topics:

1-The endolysosomal system

1.1 Lysosomes

1.2. Endocytosis and the endolysosomal system

1.3. Lysosomes as signaling hubs: from lysosomal exocytosis to autophagy control

  1. How do coronaviruses deal with the endosomal system changed to How do coronaviruses subvert the endosomal system (line 280)

    2.1 Virus entry mechanisms

          2.1.1 Fusion and endocytosis are ways SARS-CoV-2 uses to invade cells changed to Fusion and endocytosis can take  SARS-CoV2 into cells (line 396)

   2.2 Replication and assembly of Coronaviruses

   2.3 Lysosomes: an unusual way Coronaviruses use to exit the cell changed to Exit from the cell: lysosomes as alternative routes (line 634)

  1. The role of the endolysosomal system revealed by genome-wide studies changed to Genome-wide studies reveal components of the endolysosomal system as potential pharmacological targets against COVID-19 (line 826-827)
  2. Conclusion

The authors are making comparisons between SARS-CoV and SARS-CoV-2 often, but sometimes it is hard to see the relevance to the topic.

Answer: Thank you for the comment. Since several viruses use the endolysosomal system during their replication and assembly, our idea here was not only to state the general features of coronaviruses and their uses of the endolysosomal system but also to highlight the specific aspects of each virus biology since these could be used specifically for drug development against each virus type.

PRRAR sequence can also be a heparin binding domain, and RBD/heparan-sulphate interaction also may have a role in the cell entry mechanism of the virus (10.1016/j.cell.2020.09.033, 10.3390/ijms22105336).

Answer: These were included in the main text (line 420). Thank you for pointing that out.

Institutional Review Board Statement: delete text and state “not applicable”.

Answer: This will also be corrected.

Summary of alterations in the text:

Lines 121-123: These hydrolases may also be secreted by the cell through the secretory pathway and endocytosed back after they bind to mannose-6-phosphatase receptors (M6PR) present in the plasma membrane [15].

Line 280: How do coronaviruses subvert the endosomal system

Line 396: Fusion and endocytosis can take SARS-CoV2 into cells.

Line 420: A reference was included (Claussen et al., 2020).

Line 445—451: (A) Entry by fusion - the direct fusion of the SARS-CoV2 envelope with the plasma membrane is mediated by the processing of the spike protein after the interaction with the cellular receptor ACE2. The cleavage sites by specific proteases are shown: furin cleaves spike a multibasic motif in the S1/S2 site (priming step), and TMPRSS2 cleaves spike in the S2’ site (activation step). Cleavage by furin seems to occur during the cell exit of SARS-CoV-2 virions. When cells express TMPRSS2, hidden hydrophobic residues of the S2 region are exposed, and SARS-CoV-2 fuse directly with the plasma membrane.

Lines 508-512: HEK293T cells express shallow levels of ACE2 or TMPRSS2 compared to other cells (Sherman & Emmer, 2021); hence exogenous expression is usually performed to assure SARS-CoV-2 infection. Bayati and coworkers using HEK293T-ACE2hi and a spike-pseudotyped lentivirus demonstrated that infection occurred by clathrin-dependent endocytosis, inhibited by a clathrin heavy chain knockdown approach.

Line 634: Exit from the cell: lysosomes as alternative routes.

Lines 798-806: Liu and coworkers further demonstrated another mechanism of subversion of autophagy by coronaviruses. They demonstrated that infection with the coronaviruses OC43 and 229E promote TFEB degradation by activating the PAK2 kinase that phosphorylates TFEB and primes it for ubiquitin-driven degradation mediated by the E3 ubiquitin ligase subunit DCAF7. The authors identified the agents BC18813 and BC18630 that can interfere with TFEB-DCAF7 interaction. Importantly, these agents attenuated the in vitro infection of Beas-2B cells with OC43 or 229E, MCDK cells with Influenza H1N1, and Calu-3 cells with SARS-CoV-2 while restoring lysosomal function. Finally, BC18630 was also effective in attenuating SARS-CoV-2 infection in vivo using a hamster model (Liu et al., 2021).

Lines 825-826: Genome-wide studies reveal components of the endolysosomal system as potential pharmacological targets against COVID-19.

Lines 937-951: Summed to that, the use of nanotechnology to optimize drug delivery and efficacy represents a powerful tool against COVID-19 (for a review, see Seyfoori et al., 2021). Niclosamide, an approved anti-helminthic drug that stimulates autophagy, has broad antiviral activity but low oral bioavailability (Jeon et al., 2020). Importantly, it has antiviral activity against Alpha (B.1.1.7), Beta (B.1.351), and Delta (B.1617.2) SARS-CoV-2 variants in human primary lung epithelial cells, as well as in Vero and Caco-2 cell lines (Weiss et al., 2021). Nanotechnology has been, used to improve the delivery of niclosamide to treat conditions, such as cancer and now COVID-19 (Schweizer et al, 2018, Brunaugh et al., 2021). With this purpose, the study of Brunaugh and coworkers report the development of an inhalable formulation of niclosamide associated with lysozyme. Lysozyme helps the solubilization and delivery  of niclosamide  to the primary sites of infection of coronaviruses at high dose concentrations and improved the antiviral potency of niclosamide in in vitro and  in vivo MERS-CoV and SARS-CoV-2 infection using a murine model (Brunaugh et al., 2021). Inhaled and intranasal application of niclosamide was tested in a Phase I trial that revealed its use is safe, well-tolerated, and showed mild adverse effects, the most common a transient irritation of the upper airways (Backer et al., 2021).

Lines 952-955: While SARS-CoV2 keeps surprising us with new variants, scientists try to learn and understand its biology better to control its spread around the globe. We hope the knowledge of viral strategies of infection associated with the development of vaccines and the advance of new pharmacological approaches will grant us the end this pandemic soon.

Reviewer 2 Report

First of all, the authors should be well-commended and appreciated for their effort in generating such informative and interesting review article. Intracellular lysosomal degradation pathway such as autophagy has long been known to serve as an indirect escape route for invading particles of various types of viruses, hence protecting them from the host antiviral immune response.  The authors primarily focused on discussing the roles of host endolysosomal system in regulating SARS-CoV-2 replication in infected cells. While the authors have provided a detailed review on this topic, there are a few points which should be taken note of by the authors to further improve this manuscript:

1) Line 517: The authors should be very careful in their statement that HEK293T cells are not susceptible to SARS-CoV-2 infections. Though it is true that ACE2 expression levels in HEK293T cells are limited, several studies have shown that modest, yet significant SARS-CoV-2 replication could be detected using WT HEK293T cells, with low levels of viral nucleocapsid proteins being produced in infected 293T cells. The authors can refer to the following published articles:

Chu, Hin, et al. "Comparative tropism, replication kinetics, and cell damage profiling of SARS-CoV-2 and SARS-CoV with implications for clinical manifestations, transmissibility, and laboratory studies of COVID-19: an observational study." The Lancet Microbe 1.1 (2020): e14-e23.

Harcourt, Jennifer, et al. "Severe acute respiratory syndrome coronavirus 2 from patient with coronavirus disease, United States." Emerging infectious diseases 26.6 (2020): 1266.

2) References cited in the text occasionally appeared in different styles at the end of the sentence (Lines 146, 342, 587, 611, 826, 897).

3) Presentation of scientific nomenclatures in terms of capitalizing and italicizing, across the manuscript should be done in a consistent manner. (Eg. LAMP1 in the text is presented as Lamp1 in the figures; being consistent in capitalizing the first letter of Caveolin-1 in the paragraphs of page 8;  in vitro needs to be italicized (Line 648).

4) Some typing errors are detected: "and" in Line 50, "HEK193" in Line 593, "CRISP" in Line 859".

5) At the end of the manuscript, the authors mentioned about an interesting fact that development of drugs targeting the host endolysosomal system would provide a better alternative to the currently highly discussed antiviral drugs that specifically target the different components of SARS-CoV-2. The authors should discuss further on this by providing some insightful scientific suggestions with citations of previous papers involving different drug delivery technologies (Eg. nanoparticles), on how to better utilize the knowledge and information that have been reviewed in this manuscript for the improvement on drug discovery for SARS-CoV-2 or coronaviruses as a whole.

Author Response

Reviewer 2

Comments and Suggestions for Authors

First of all, the authors should be well-commended and appreciated for their effort in generating such informative and interesting review article. Intracellular lysosomal degradation pathway such as autophagy has long been known to serve as an indirect escape route for invading particles of various types of viruses, hence protecting them from the host antiviral immune response.  The authors primarily focused on discussing the roles of host endolysosomal system in regulating SARS-CoV-2 replication in infected cells. While the authors have provided a detailed review on this topic, there are a few points which should be taken note of by the authors to further improve this manuscript:

Answer: Thank you very much for your kind words. We really appreciate you taking the time to provide us with a careful and insightful review of our work.

1) Line 517: The authors should be very careful in their statement that HEK293T cells are not susceptible to SARS-CoV-2 infections. Though it is true that ACE2 expression levels in HEK293T cells are limited, several studies have shown that modest, yet significant SARS-CoV-2 replication could be detected using WT HEK293T cells, with low levels of viral nucleocapsid proteins being produced in infected 293T cells. The authors can refer to the following published articles:

Chu, Hin, et al. "Comparative tropism, replication kinetics, and cell damage profiling of SARS-CoV-2 and SARS-CoV with implications for clinical manifestations, transmissibility, and laboratory studies of COVID-19: an observational study." The Lancet Microbe 1.1 (2020): e14-e23.

Harcourt, Jennifer, et al. "Severe acute respiratory syndrome coronavirus 2 from patient with coronavirus disease, United States." Emerging infectious diseases 26.6 (2020): 1266.

Answer: Thank you for your criticism. We have adjusted the text to give precise information (Lines 508-512).

2) References cited in the text occasionally appeared in different styles at the end of the sentence (Lines 146, 341, 587, 611, 826, 897).

Answer: Thank you for your careful reading. These errors were corrected (lines 142, 337, 578, 602, 824, 895).

3) Presentation of scientific nomenclatures in terms of capitalizing and italicizing, across the manuscript should be done in a consistent manner. (Eg. LAMP1 in the text is presented as Lamp1 in the figures; being consistent in capitalizing the first letter of Caveolin-1 in the paragraphs of page 8;  in vitro needs to be italicized (Line 648).

Answer: Thank you for your careful reading. These errors were corrected (line 332, figures 1 and 3, line 639).

4) Some typing errors are detected: "and" in Line 50, "HEK193" in Line 593, "CRISP" in Line 859".

Answer: Thank you for your careful reading. These errors were corrected (lines 49, 584, 857).

5) At the end of the manuscript, the authors mentioned about an interesting fact that development of drugs targeting the host endolysosomal system would provide a better alternative to the currently highly discussed antiviral drugs that specifically target the different components of SARS-CoV-2. The authors should discuss further on this by providing some insightful scientific suggestions with citations of previous papers involving different drug delivery technologies (Eg. nanoparticles), on how to better utilize the knowledge and information that have been reviewed in this manuscript for the improvement on drug discovery for SARS-CoV-2 or coronaviruses as a whole.

Answer: Thank you for your criticism and valuable suggestions. We have included a text in the conclusion section describing the use of niclosamide as a very promising drug against COVID-19, commenting on recent studies on the development of formulations of this drug with nanoparticles, which help increase the delivery to the primary sites of infection, and the drug antiviral potency. We really appreciate your suggestion. We believe it helped improve the quality of our work. (see lines 937-951)

We also added information about a recent work showing another mechanism of subversion of autophagy by coronaviruses and the identification of drugs able to attenuate SARS-COV-2 infection both in vitro and in vivo (Liu et al., 2021) – (see lines 798-806).

At the end of the manuscript, we also added a short text to tie things up (see lines 952-955).

Summary of alterations in the text:

Lines 121-123: These hydrolases may also be secreted by the cell through the secretory pathway and endocytosed back after they bind to mannose-6-phosphatase receptors (M6PR) present in the plasma membrane [15].

Line 280: How do coronaviruses subvert the endosomal system

Line 396: Fusion and endocytosis can take SARS-CoV2 into cells.

Line 420: A reference was included (Claussen et al., 2020).

Line 445—451: (A) Entry by fusion - the direct fusion of the SARS-CoV2 envelope with the plasma membrane is mediated by the processing of the spike protein after the interaction with the cellular receptor ACE2. The cleavage sites by specific proteases are shown: furin cleaves spike a multibasic motif in the S1/S2 site (priming step), and TMPRSS2 cleaves spike in the S2’ site (activation step). Cleavage by furin seems to occur during the cell exit of SARS-CoV-2 virions. When cells express TMPRSS2, hidden hydrophobic residues of the S2 region are exposed, and SARS-CoV-2 fuse directly with the plasma membrane.

Lines 508-512: HEK293T cells express shallow levels of ACE2 or TMPRSS2 compared to other cells (Sherman & Emmer, 2021); hence exogenous expression is usually performed to assure SARS-CoV-2 infection. Bayati and coworkers using HEK293T-ACE2hi and a spike-pseudotyped lentivirus demonstrated that infection occurred by clathrin-dependent endocytosis, inhibited by a clathrin heavy chain knockdown approach.

Line 634: Exit from the cell: lysosomes as alternative routes.

Lines 798-806: Liu and coworkers further demonstrated another mechanism of subversion of autophagy by coronaviruses. They demonstrated that infection with the coronaviruses OC43 and 229E promote TFEB degradation by activating the PAK2 kinase that phosphorylates TFEB and primes it for ubiquitin-driven degradation mediated by the E3 ubiquitin ligase subunit DCAF7. The authors identified the agents BC18813 and BC18630 that can interfere with TFEB-DCAF7 interaction. Importantly, these agents attenuated the in vitro infection of Beas-2B cells with OC43 or 229E, MCDK cells with Influenza H1N1, and Calu-3 cells with SARS-CoV-2 while restoring lysosomal function. Finally, BC18630 was also effective in attenuating SARS-CoV-2 infection in vivo using a hamster model (Liu et al., 2021).

Lines 825-826: Genome-wide studies reveal components of the endolysosomal system as potential pharmacological targets against COVID-19.

Lines 937-951: Summed to that, the use of nanotechnology to optimize drug delivery and efficacy represents a powerful tool against COVID-19 (for a review, see Seyfoori et al., 2021). Niclosamide, an approved anti-helminthic drug that stimulates autophagy, has broad antiviral activity but low oral bioavailability (Jeon et al., 2020). Importantly, it has antiviral activity against Alpha (B.1.1.7), Beta (B.1.351), and Delta (B.1617.2) SARS-CoV-2 variants in human primary lung epithelial cells, as well as in Vero and Caco-2 cell lines (Weiss et al., 2021). Nanotechnology has been, used to improve the delivery of niclosamide to treat conditions, such as cancer and now COVID-19 (Schweizer et al, 2018, Brunaugh et al., 2021). With this purpose, the study of Brunaugh and coworkers report the development of an inhalable formulation of niclosamide associated with lysozyme. Lysozyme helps the solubilization and delivery  of niclosamide  to the primary sites of infection of coronaviruses at high dose concentrations and improved the antiviral potency of niclosamide in in vitro and  in vivo MERS-CoV and SARS-CoV-2 infection using a murine model (Brunaugh et al., 2021). Inhaled and intranasal application of niclosamide was tested in a Phase I trial that revealed its use is safe, well-tolerated, and showed mild adverse effects, the most common a transient irritation of the upper airways (Backer et al., 2021).

Lines 952-955: While SARS-CoV2 keeps surprising us with new variants, scientists try to learn and understand its biology better to control its spread around the globe. We hope the knowledge of viral strategies of infection associated with the development of vaccines and the advance of new pharmacological approaches will grant us the end this pandemic soon.

Round 2

Reviewer 1 Report

I have no further comments to the authors.